# Hyperedge2vec: Distributed Representations for Hyperedges

## Abstract

Data structured in form of overlapping or non-overlapping sets is found in a variety of domains, sometimes explicitly but often subtly. For example, teams, which are of prime importance in social science studies are "sets of individuals"; "item sets" in pattern mining are sets; and for various types of analysis in language studies a sentence can be considered as a "set or bag of words". Although building models and inference algorithms for structured data has been an important task in the fields of machine learning and statistics, research on "set-like" data still remains less explored. Relationships between pairs of elements can be modeled as edges in a graph. However, modeling relationships that involve all members of a set, a hyperedge is a more natural representation for the set. In this work, we focus on the problem of embedding hyperedges in a hypergraph (a network of overlapping sets) to a low dimensional vector space. We propose a probabilistic deep-learning based method as well as a tensor-based algebraic model, both of which capture the hypergraph structure in a principled manner without loosing set-level information. Our central focus is to highlight the connection between hypergraphs (topology), tensors (algebra) and probabilistic models. We present a number of interesting baselines, some of which adapt existing node-level embedding models to the hyperedge-level, as well as sequence based language techniques which are adapted for set structured hypergraph topology. The performance is evaluated with a network of social groups and a network of word phrases. Our experiments show that accuracy wise our methods perform similar to those of baselines which are not designed for hypergraphs. Moreover, our tensor based method is quiet efficient as compared to deep-learning based auto-encoder method. We therefore, argue that we have proposed more general methods which are suited for hypergraphs (and therefore also for graphs) while maintaining accuracy and efficiency.

## 1 Introduction

In group structured data we have multiple entities related by some form of group relationships. In fact, such data is more abundantly found in the real world than has been usually studied (Estrada & Rodriguez-Velazquez, 2005). There are increasing number of fields where such data is being found more than ever before. In social networks domain: team data from massive online multi-player games (Ahmed et al., 2011) such as World of Warcraft, group communication tools such as Skype and Google Docs and research collaborations (Subbian et al., 2013; pub). There are other fields in which structural relationships between entities is important as well, and large datasets capturing them exist. Examples include Natural Language Processing (Bengio & Bengio, 2000), Biology (Hwang et al., 2008; Klamt et al., 2009), e-commerce (Deshpande & Karypis, 2004; Christakopoulou & Karypis, 2014) and Chemistry (Bartholomay, 1960). Figure 1 shows such examples for networks of groups, sentences, and item sets.

Hypergraph (Berge, 1984), which is a generalization of graphs, is a popular model to naturally capture higher-order relationships between sets of objects (Figure 2) (Estrada & Rodriguez-Velazquez, 2005). Within machine learning, algorithms guided by the structure of such higher order networks (Zhou et al., 2006) have found applications in a variety of domains (Tian et al., 2009; Gao et al., 2013; Li & Li, 2013; Sharma et al., 2015; 2017).

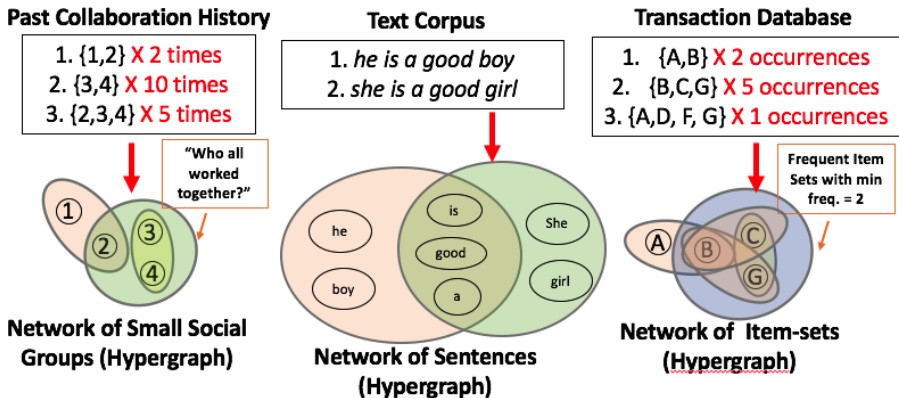

Figure 1: Example illustrating "set-like" structures from various domains.

More recently, the interest of representation learning in NLP (Mikolov et al., 2013a) has stimulated its application in graph embedding (learning low dimensional representation for graph nodes) (Perozzi et al., 2014; Tang et al., 2015b; Grover & Leskovec, 2016; Cai et al., 2017). However, this new line of research is limited to simple graphs and we are unaware of any work that considers hyperedge embeddings (learning representations for hyperedges). In this paper we aim to develop methods which (1) learn hypergraph embeddings directly, (2) leverage the hypergraph topology and (3) not loose the hyperedge-level joint information. These learned embeddings can then be employed by a supervised or semi-supervised algorithm to perform various predictive tasks pertaining to hyperedges. For example performance prediction of a team (set of individuals) engaged in a collaborative task or sentiment analysis of a sentence (set of words).

Although, there have been a variety of attempts to learn *node* embeddings for hypergraphs by extending traditional graph embedding methods for hypergraph setting. These approaches differ in the extent they address the aims (2) and (3) as pointed previously. In first category, there are a number of methods that incorporate the hypergraph topology using proxy graphs (Zhou et al., 2006; Hwang et al., 2008), therefore, incur loss of information. Also, Agarwal et al. (2006) squarely criticize that such representations can be learned by constructing graphs, which are proxies for the hypergraph structure. Second, are methods which actually, take into account the hyperedge-level information, like the classical Neural Language Model which maintains the higher order sequence information (Bengio & Bengio, 2000). But their model is designed for sequences and also simply ignores the topology (sequences are linked by common words forming a network of sequences). However recently, there has been interest in modeling "set-like" structures within deep-learning community (Vinyals et al., 2016; Rezatofighi et al., 2016). But they do not consider the hypergraph structure between the sets and therefore, not model hypergraphs in a principled manner. Third category are those methods that do not ignore the topology completely and also retain the hyperedge-level information. Among this are $k$-way tensor based methods that explicitly do not work on hypergraph (Shashua et al., 2006; Bulò & Pelillo, 2009), but the connection exists in the form that $k$-way tensor represents a k-uniform hypergraph (Qi, 2005). But they are restricted to uniform hypergraphs.

We therefore, propose two methods, both of which directly learn hyperedge embeddings for general hypergraphs and capture the hypergraph structure in a principled manner. First, is an auto-encoder based deep-learning method and second, is a tensor-based algebraic method. We also highlight the connection between hypergraphs (topology), tensors (algebra) and probabilistic models. A number of interesting baselines are also developed, some of which adapt existing node-level embedding models to hypergraph setting and also sequence based NLP techniques are adapted for set structured hypergraphs. We have experimented with social group dataset (network of teams as a hypergraph) from an online game for team (hyperedge) performance prediction as well as language networks (sentence/phrase hypergraphs) for phrase-level sentiment analysis. Our experiments show that accuracy wise our methods perform similar to those of baselines which are not designed for hypergraphs. Moreover, our tensor based method is quiet efficient as compared to deep-learning based auto-encoder method. We therefore, argue that we have proposed more general methods which are suited for hypergraphs (and therefore also for graphs) while maintaining accuracy and efficiency. In summary the main contributions of our paper are:

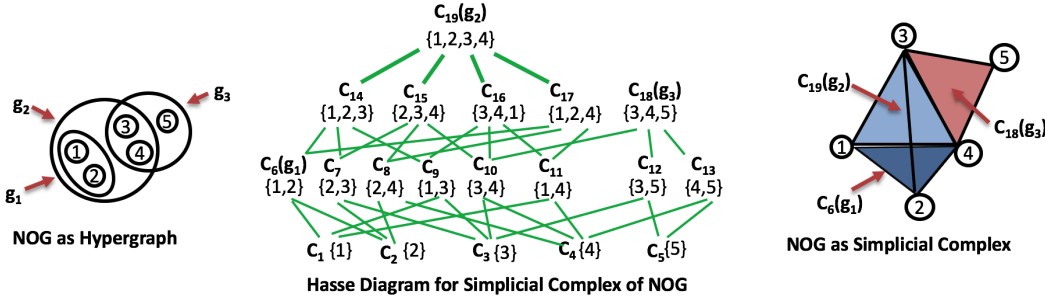

Figure 2: Example illustrating *network of groups (NOG)* which is the *hypergraph (HG)* (top left), the corresponding *simplicial complex (SC)* (top right) and the *hasse diagram (HD)* (bottom) corresponding to the simplicial complex, for a scenario where the actors $\{1,2,3,4,5\}$ have collaborated in past as groups: $g_1 = \{1, 2\}$, $g_2 = \{1, 2, 3, 4\}$ and $g_3 = \{3, 4, 5\}$.

- We propose the novel concept of a *dual tensor* corresponding to the hypergraph dual and allows us to get hyperedge embedding directly.

- We propose a general *hypergraph tensor decomposition* method designed for general hypergraphs (containing different cardinality hyperedges) unlike simple uniform hypergraph tensor decomposition which is restricted to fixed cardinality hyperedges (i.e. uniform hypergraph). We are unaware of any such works or applications employing this approach.

- We propose the novel idea to use of de-noising auto-encoder in a hypergraph setting. Moreover, we also develop techniques of creating noise using random-walks over hasse diagram topology, which is original and unique.

- We highlight and argue that embeddings from tensor based methods have a natural hypergraph theoretic interpretation unlike the deep learning based black-box approaches.

- Our proposed methods learn hypergraph embeddings based on the unique approach of leveraging the *existing structure* present in network data as context, in contrast to context generation based existing graph embedding techniques.

Following is the outline for the rest of the paper. In section 2 we describe the problem definition and statement followed by Section 3 where we describe in detail the various methods proposed in this paper. Section 4 describes the datasets, experimental tasks & settings for the models. Section 5 provides an overview of the related literature followed by conclusion and appendix.

## 2 PRELIMINARIES

In this paper we consider the scenario where we have a collection of *elements*. These elements can represent individual actors in case of social groups or words in sentences or items in item-sets within a transaction database. In other words a social group or a sentence or an item-set are *sets* which contain these *elements*. Let $V = \{v_1, v_2, ..., v_n\}$ represents $n$ elements and we have $m$ sets defined over these elements, denoted by $G = \{g_1, g_2, ..., g_m\}$, where $g_i \subseteq V$ represents the $i^{th}$ set. The cardinality $|g_i|$ represents the number of elements in the set. Also each set $g_i \in G$ has an occurrence number $R(g_i)$, which denotes the number of times it has occurred. Such overlapping or non-overlapping sets can be modeled as a *hypergraph* Berge (1984), where the nodes and hyperedges, represent the elements and sets, respectively. This hypergraph is represented as $N_g = (V, G)$ with $G$ as the collection of hyperedges over the nodes $V$. The incidence matrix $\mathbf{H} \in \{0, 1\}^{|G| \times |V|}$ for $N_g$ represents the presence of nodes in different hyperedges with $\mathbf{H}(g_i, v) = 1$ if $v \in g_i$ else 0. We also define degree $d(v)$ of a vertex $v$ as the number of hyperedges incident on this vertex i.e. $d(v) = \sum_{g_i \in G} \mathbf{H}(g_i, v)$. A specialization of hypergraph model is the *simplicial complex* (Munkres, 1984) (Figure 2), in which additionally each hyperedge has the subset closure property, i.e., each subset of hyperedge is also a valid hyperedge.

**Problem Statement:** Given this setting, our goal is to learn the mapping $\phi : G \to \mathbb{R}^d$ from hyperedges to feature representations (i.e., embeddings) that can be used to build predictive models

involving sets. Here $d$ is a parameter specifying the number of dimensions of the embedding vector. Equivalently, $\phi$ can be thought of as a look-up matrix of size $|G| \times d$, where $|G|$ is the total number of sets or hyperedges.

# 3 METHODOLOGY

As we had mentioned before there are several methods to learn representation of nodes in a graph. Given that a hyperedge is a set of nodes, a natural question arises is that, can we combine the node level embeddings (learned using existing methods) within a given hyperedge to find a suitable representation of that hyperedge? However, there are large number of possible ways one can combine the node embeddings. Therefore, in the following two subsections, we develop two methods that learn the embeddings for hyperedges directly in a more principled manner. Our tensor based method is focused on retaining the set-level information intact while harnessing the hypergraph network structure. Where the auto-encoder based method harnesses the non-linearity of representation offered by deep learning to generate crisp and powerful embeddings. Tensor based method has a natural *hasse diagram* based topological interpretation and can generate both node and hyperedge embeddings. Whereas deep learning model is more black-box based approach and can only generate hyperedge embeddings.

## 3.1 HYPEREDGE2VEC USING HYPERGRAPH TENSOR DECOMPOSITION

In this section, we develop tensor (higher-order matrices) based linear algebraic methods that learn node as well as hyperedge embedding by taking into account the joint probability over a hyperedge. The idea behind using tensors is that they retain the set-level information contained in a hypergraph, unlike the proxy graphs (corresponding to hypergraphs) based techniques (used as baselines in our experiments), which approximate hyperedge or set-level information with dyadic edge-level information. Therefore, we design an algorithm which is principally suited for hypergraph structured data. For a given hypergraph we can extract a sub-hypergraph that only consists of the hyperedges with cardinality $k$. This sub-hypergraph is a $k$-uniform hypergraph or $k$-graph (Cooper & Dutle, 2012). Corresponding to this $k$-uniform hypergraph, we can define a $k^{th}$ order $n$-dimensional symmetric tensor (Qi, 2005) $\mathcal{A}^{\mathbf{k}}_{\mathbf{hyp}} = (a_{p_1,p_2,..,p_k}) \in \mathbb{R}^{[k,n]}$ whose elements are initialized as follows:

$$a_{p_1,p_2,..,p_k} = R(g_i) \tag{1}$$

where $\{v_{p_1}, v_{p_2}, ..., v_{p_k}\} \in g_i$ and $|g_i| = k, \forall i \in \{1, ..., m\}$. Note that symmetry here implies that value of element $a_{p_1,p_2,..,p_k}$ is invariable under any permutation of its indices $(p_1, p_2, .., p_k)$. Rest all the elements in the tensor are zeros.

In a similar manner we can also define a *dual tensor*, corresponding to *hypergraph dual* where the roles of nodes and hyperedges are interchanged. We consider all the hyperedges in the hypergraph dual that are of cardinality $k$. This basically corresponds to all the vertices in the original hypergraph which have a degree of $k$, i.e., they are part of exactly $k$ hyperedges in the original hypergraph. Corresponding to this $k$-uniform hypergraph dual, we can define a $k^{th}$ order $m$-dimensional symmetric dual tensor $\mathcal{A}^{\mathbf{k}}_{\mathbf{dual}} = (a_{q_1,q_2,..,q_k}) \in \mathbb{R}^{[k,m]}$ whose elements are initialized as follows:

$$a_{q_1,q_2,..,q_k} = 1 \tag{2}$$

where $\{g_{q_1}, g_{q_2}, ..., g_{q_k}\} \ni v_j$ and $d(v_j) = k, \forall j \in \{1, ..., n\}$. Note that this tensor is also symmetric and rest all the elements in the tensor are zeros. Moreover, we can also have other meaningful initialization schemes, for that we refer to Appendix B.1. To realize our aim of learning node and hyperedge embeddings we perform joint CP Tensor Decomposition (Kolda & Bader, 2009) (of the tensors we just described) across different cardinality hyperedges simultaneously. Specifically, for the node embeddings we solve the following optimization problem:

$$\min_{\mathbf{Z}} \sum_{k=c_{min}}^{c_{max}} \|\mathcal{A}^{\mathbf{k}}_{\mathbf{hyp}} - \widehat{\mathcal{A}^{\mathbf{k}}_{\mathbf{hyp}}}\| \tag{3}$$

where,

$$\widehat{\mathcal{A}^{\mathbf{k}}_{\mathbf{hyp}}} = \sum_{r=1}^{d} \lambda_r \underbrace{\mathbf{z}_r \circ \mathbf{z}_r ... \circ \mathbf{z}_r}_{k \text{ times}} \equiv \langle \lambda | \underbrace{\mathbf{Z}, \mathbf{Z}, ...., \mathbf{Z}}_{k \text{ times}} \rangle \tag{4}$$

with $\mathbf{z}_r \in \mathbb{R}^n$, $\lambda_r \in \mathbb{R}_+$, $\mathbf{Z} \in \mathbb{R}^{n \times d}$ and $\mathbf{Z}(:,r) = \mathbf{z}_r$. Notice, that equation 4 is the standard *symmetric* CP decomposition but equation 3 is summation of reconstruction errors in different tensor decompositions for different cardinality hyperedges. Each error term learns a common latent factor matrix $\mathbf{Z}$ using the empirical observed $k$-uniform sub-hypergraph stored in $\mathcal{A}^{\mathbf{k}}_{\mathbf{hyp}}$ tensor. We perform this joint decomposition by augmenting the standard CP Decomposition into the **Hypergraph-CP-ALS** algorithm 1. Lines (4-8) is the standard (*non-symmetric*) CP Decomposition (see reference Kolda & Bader (2009) for details), which solves the following optimization:

$$\widehat{\mathcal{A}^{\mathbf{k}}_{\mathbf{hyp}}} = \sum_{r=1}^{d} \lambda_r \, {}_k\mathbf{u}_r^{(1)} \circ {}_k\mathbf{u}_r^{(2)} ... \circ {}_k\mathbf{u}_r^{(k)} \equiv \langle \lambda | {}_k\mathbf{U}^{(\mathbf{1})}, {}_k\mathbf{U}^{(\mathbf{2})}, ...., {}_k\mathbf{U}^{(\mathbf{k})} \rangle \qquad (5)$$

with ${}_k\mathbf{u}_r^{(j)} \in \mathbb{R}^n$, $\lambda_r \in \mathbb{R}_+$, ${}_k\mathbf{U}^{(\mathbf{j})} \in \mathbb{R}^{n \times d}$, ${}_k\mathbf{U}^{(\mathbf{j})}(:,r) = {}_k\mathbf{u}_r^{(j)}$ and $j \in \{1,..,k\}$. As we want to learn common representations ($\mathbf{Z}$) for all the nodes, we add additional constraint that enforce that ${}_k\mathbf{U}^{(\mathbf{j})}$ are same for all $j \in \{1,..,k\}$ and for all $k \in [c_{min}, c_{max}]$. Lines (9-15) is where we force all representations for all cardinalities to be average of the ${}_k\mathbf{U}^{(\mathbf{j})}$ achieved by the last decomposition iteration that occurred in lines (4-8). Averaging can be interpreted as equal contribution from the latent factors learned from different cardinality (uniform) sub-hypergraphs. We make repeated pass through the entire hypergraph (by learning via different $k$-uniform sub-hypergraph (line 3)) until the objective (equation 3) converges. In our implementation we empirically observe that using the above mentioned unbiased averaging heuristic, our algorithm converges successfully. The same algorithm 1 is used to get the hyperedge embeddings, by just passing $\mathcal{A}_{\mathbf{dual}}$ instead of $\mathcal{A}_{\mathbf{hyp}}$. We shall jointly refer to the embeddings achieved for nodes and hyperedges via the above tensor decomposition techniques as **t2v**.

---

**Algorithm 1 Hypergraph-CP-ALS** $(\mathcal{A}_{\mathbf{hyp}}, c_{min}, c_{max})$

---

1: randomly initialize ${}_k\mathbf{U}^{(\mathbf{j})}, \forall k \in [c_{min}, c_{max}], \forall j \in \{1, ..., k\}$
2: **repeat**
3:     **for** $k = c_{min}$ to $c_{max}$ **do**
4:         **for** $j = 1$ to $k$ **do**
5:             $\mathbf{V} \leftarrow {}_k\mathbf{U}^{(\mathbf{1})\mathsf{T}} {}_k\mathbf{U}^{(\mathbf{1})} * ... * {}_k\mathbf{U}^{(\mathbf{j-1})\mathsf{T}} {}_k\mathbf{U}^{(\mathbf{j-1})} * {}_k\mathbf{U}^{(\mathbf{j+1})\mathsf{T}} {}_k\mathbf{U}^{(\mathbf{j+1})} * ... * {}_k\mathbf{U}^{(\mathbf{j})\mathsf{T}} {}_k\mathbf{U}^{(\mathbf{j})}$
6:             ${}_k\mathbf{U}^{(\mathbf{j})} \leftarrow (\mathcal{A}^{\mathbf{k}}_{\mathbf{hyp}})^{(j)} ({}_k\mathbf{U}^{(\mathbf{k})} \odot ... \odot {}_k\mathbf{U}^{(\mathbf{j+1})} \odot {}_k\mathbf{U}^{(\mathbf{j-1})} \odot ... \odot {}_k\mathbf{U}^{(\mathbf{1})}) \mathbf{V}^\dagger$
7:             normalize columns of ${}_k\mathbf{U}^{(\mathbf{j})}$ (and store norms as $\lambda$)
8:         **end for**
9:         $\mathbf{Z} \leftarrow \frac{1}{k} \sum_{j=1}^{k} {}_k\mathbf{U}^{(\mathbf{j})}$
10:         **for** $p = c_{min}$ to $c_{max}$ **do**
11:             **for** $j = 1$ to $p$ **do**
12:                 ${}_p\mathbf{U}^{(\mathbf{j})} \leftarrow \mathbf{Z}$
13:             **end for**
14:         **end for**
15:     **end for**
16: **until** fit criteria achieved or maximum number of iterations exceeded
17: **return** $\mathbf{Z}$

---

We would like to highlight a few points regarding the tensor methods. The tensors that we have employed are super-symmetric and hence able to capture distribution over sets rather than sequence. But in general we can employ a $k$-way tensor which is not symmetric to even capture sequence. In this sense tensors are more general purpose. Another point one can observe that when we initialize the hypergraph tensor $\mathcal{A}^{\mathbf{k}}_{\mathbf{hyp}}$, we have initialized all the permutations of vertices corresponding to a given hyperedge (Eq. 26). Moreover, we initialize it by the repetition counts.

## 3.2 Hyperedge2Vec Using Hasse De-noising Autoencoder

An autoencoder (Bengio et al., 2009), takes an input vector $\mathbf{x} \in [0,1]^n$ and maps it to a latent representation $\mathbf{y} \in [0,1]^d$. This is typically done using an affine mapping followed by a non-linearity (more so when the input, like in our case, is binary (Vincent et al., 2010)): $f_{en}(\mathbf{x}) = s(\mathbf{Wx} + \mathbf{b})$, with parameters $\theta = \{\mathbf{W}, \mathbf{b}\}$. Here, $\mathbf{W}$ is a $n \times d$ weight matrix and $\mathbf{b}$ is the offset.

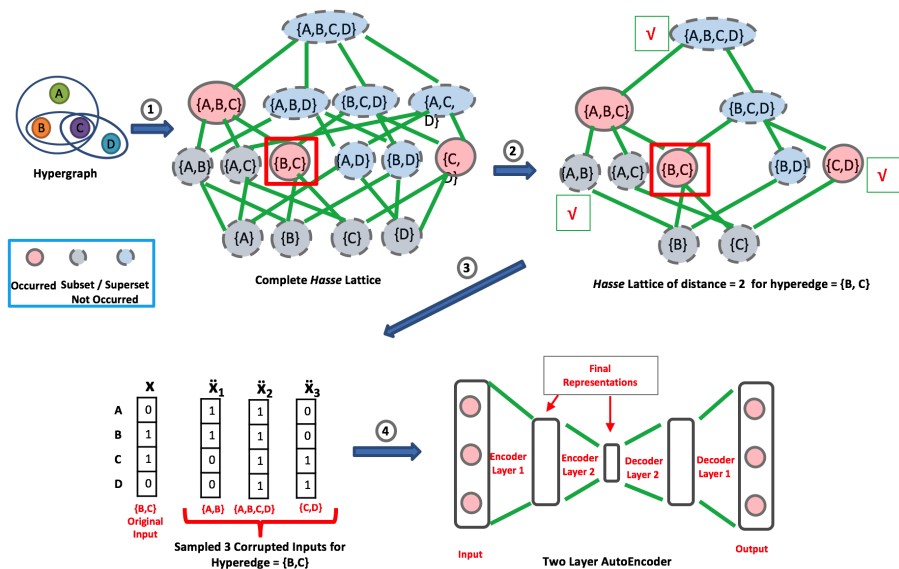

Figure 3: For the given hypergraph between four nodes (A,B,C,D) we consider the complete *hasse* lattice. For a given hyperedge {B,C} (square box) we then construct the sub-lattice made of hyperedges with distance $h = 2$ from {B,C}. We perform random walk (with $\tau = 0.2$) starting from the node corresponding to hyperedge {B,C} and sample $p = 3$ hyperedges (nodes visited by the random walk; shown with a check-mark ✓). Finally, we train the autoencoder to reconstruct the original hyperedge from these $p$ noisy hyperedges.

This latent representation is reconstructed into a vector $\mathbf{z} = f_{de}(\mathbf{y}) = s(\mathbf{W}'\mathbf{y} + \mathbf{b}')$, in the input space, $\mathbf{z} \in [0,1]^n$ and parameters $\theta' = \{\mathbf{W}', \mathbf{b}'\}$. The mappings $f_{en}$ and $f_{de}$ are referred as the **encoder** and **decoder**, respectively. The representation $\mathbf{y}$ is learned by minimizing the following reconstruction error:

$$\theta^*, \theta'^* = \arg\min_{\theta^*, \theta'^*} \frac{1}{m} \sum_{i=1}^{m} L(\mathbf{x}_i, \mathbf{z}_i) = \arg\min_{\theta^*, \theta'^*} \frac{1}{m} \sum_{i=1}^{m} L(\mathbf{x}_i, g_{\theta'}(f_\theta(\mathbf{x}_i))) \tag{6}$$

where $L$ is a loss function, which in case of binary or bit probabilities is often chosen as the *cross-entropy loss*:

$$L(\mathbf{x}, \mathbf{z}) = \sum_{j=1}^{n} [\mathbf{x}(j) \log \mathbf{z}(j) + (1 - \mathbf{x}(j)) \log(1 - \mathbf{z}(j))] \tag{7}$$

In their paper, Vincent et al. (2010) have shown that minimizing reconstruction amounts to maximizing the lower bound on the mutual information between input ($x$) and the representation $\mathbf{y}$. However, they have further argued (Vincent et al. (2008)) that $\mathbf{y}$ retaining information about input $\mathbf{x}$ is insufficient. They further, propose the idea that the learned representation should be able to recover (*denoising*) the original input even after being trained with corrupted input (*adding noise*). They generate the corrupted input ($\tilde{\mathbf{x}}$), using a stochastic mapping $q(\tilde{\mathbf{x}}|\mathbf{x})$. Choice of noise is usually either *Gaussian* for real inputs and *Salt-and-pepper noise* for discrete inputs. The *denoising autoencoder* then learns the representation for each input, $\mathbf{x}$, same as Equation 6, but with the following modified loss function: $L(\mathbf{x}(i), g_{\theta'}(f_\theta(\tilde{\mathbf{x}}(i))))$.

We leverage the denoising autoencoder for learning representation for $j^{th}$ hyperedge, by treating each hyperedge as an input, $\mathbf{x} = \mathbf{H}(j, :)$. The size of this input vector for each hyperedge is $n$, which is the number of vertices in the hypergraph. In most natural hypergraphs, specially social networks, $n$ can be quiet high ranging from thousands to millions or even billions (like Facebook for example). Therefore, randomly using a discrete noise like *salt-and-pepper*, might not be reasonable, as there are large number of possible permutations (as size $n$ is large) and not all of them are related. Random addition of 1s or deletion of existing 1s from $\mathbf{x}$, amounts to randomly adding or deleting vertices to the hyperedge corresponding to $\mathbf{x}$. This might end up in new hyperedges that are completely unrelated to the given hyperedge ($\mathbf{x}$). For example, users (nodes) in a social network

from completely different regions of the network suddenly form a group (hyperedge). Such anomalous scenarios rarely happen in practice and social groups evolve in a gradual fashion via simple processes (Sharma et al., 2017; 2015).

Rather, we take advantage of the hypergraph structure to systematically guide us in generating this noise. A hypergraph can be defined by its corresponding *hasse lattice* (Sharma et al., 2017). For a given hyperedge ($\mathbf{x}$), we consider the sub-lattice consisting of only those hyperedges that are a distance $h$ from it in the complete lattice. On this sub-lattice we sample $p$ hyperedges (nodes in sub-lattice) by performing random walk starting at the given hyperedge's node (see Figure 3). We assume that all the nodes in the sub-lattice which correspond to previously occurred hyperedge $g_i$ have weight $R(g_i)$ and rest all nodes have a constant weight $\tau$. During random walk from a given node we choose a neighboring node (as the next node) in proportion to this neighboring node's weight as compared to the other neighboring nodes. Our stochastic mapping $q(\tilde{\mathbf{x}}|\mathbf{x})$ is therefore, a random walk on the sub-lattice of hyperdge ($\mathbf{x}$) containing hyperedges at distance $h$ from it. Intuitively, the hyperedges coming within a reasonable distance will affect each others representations and will have more similar representations. We will refer to the hyperedge representations learned by the above autoencoder technique, as **h2v-auto**.

## 4 EXPERIMENTS

### 4.1 DATASET DESCRIPTION

As the first dataset, we use group interaction log-data of the Sony's Online multi-player game EverQuest II (**EQ II**) (www.everquest2.com) for a time period of nine months. In this game, several players work together as a team to perform various tasks. Each team earn points after completion of each task, and as the teams progress by earning points, they are escalated to different levels of game play. The interestingness of the game increases with each level. The points earned by the teams are treated as a measure of group performance. Each set of players who played together is treated as a hyperedge. We treat the number of times same set of players play together again as hyperedge occurrence number ($R(g_i)$). Players can participate in several teams over time, therefore, resulting in a hypergraph with overlapping hyperedges. We consider hyperedges of cardinality $\in [2, 6]$ as almost $90\%$ of our hyperedges lie within this range. The resulting dataset contains a total of 5964 hyperedges (teams) among 6536 nodes (players).

Second dataset, is the fully labeled and publicly available sentiment analysis corpus of *Stanford Sentiment Treebank* (**LangNet**) (Socher et al., 2013). This dataset is based on the reviews from a movie review website (www.rottentomatoes.com) and contains 215,154 unique phrases. Each of the phrases are labeled with a sentiment score (a real number $\in [0, 1]$, larger value indicates positive sentiment) by human annotators. Each phrase is a set or hyperedge of words. As there is no occurrence information for a phrase hyperedge we consider $R(g_i) = 1, \forall i \in \{1, ..., m\}$. Given that words are shared across various phrases, these common words connect the phrase hyperedges, resulting in a phrase hypergraph with overlapping phrase hyperedges. Again, we only consider hyperedges of cardinality $\in [2, 6]$. After applying this cardinality filter we are left with 141,410 hyperedges (phrases) and 21,122 nodes (words).

### 4.2 EVALUATION METHODOLOGY AND EXPERIMENTAL SETUP

#### 4.2.1 METHODS COMPARED

As mentioned before we refer to our proposed methods: tensor based *hypergraph tensor decomposition* and deep auto-encoder based *hypergraph auto-encoder*, as **t2v** and **h2v-auto**, respectively. We compare our proposed methods against six baselines which are listed below. Methods (1-2) are adapted from sequence based language models for set structured hypergraph data. Methods (3-6) are various kind of dyadic graph based embedding methods adapted for hypergraph setting.

1. **h2v-DM:** Hyperedge and node embeddings obtained using sentence embedding using distributed memory (DM) model (refer section A.1)

2. **h2v-DBOW:** Hyperedge and node embeddings obtained using sentence embedding using distributed memory (DBOW) model (refer section A.1)

3. **h2v-inv:** Hyperedge and node embeddings obtained using node2vec on inverted hypergraph and simple graph, respectively (refer section A.2)

4. **h2v-dual:** Hyperedge and node embeddings obtained using node2vec on hypergraph dual and the proxy graph for hypergraph, respectively (refer section A.2)

5. **e2v:** Hyperedge and node spectral embeddings obtained using eigen decomposition of inverted hypergraph laplacian and graph laplacian, respectively (refer section A.3)

6. **e2v-hyp:** Hyperedge and node spectral embeddings obtained using eigen decomposition of dual laplacian and hypergraph laplacian, respectively (refer section A.3)

The details of these baseline can be found in Appendix A. Except for **h2v-auto**, each of the baselines as well as **t2v** outputs both node as well as hyperedge embeddings of dimension $d = 128$. We further combine the node and hyperedge embedding using five different strategies: (i) node embedding summation (dimension $d = 128$), (ii) node embedding summation and concatenation with hyperedge embedding (dimension $2 \times d = 256$), (iii) node embedding averaging (dimension $d = 128$), (iv) node embedding averaging and concatenation with hyperedge embedding (dimension $2 \times d = 256$), and (v) only hyperedge embedding (dimension $d = 128$). **h2v-auto** only produces hyperedge embeddings of dimension $d = 128$. But it builds the embeddings using three different scenarios as mentioned in next section. Therefore, in total we have $38 (= 35 + 3)$ different scenarios each resulting in a different hyperedge embedding.

### 4.2.2 Evaluation Tasks and Setup

We perform two regression based tasks for the two datasets. In **EQII** dataset each team (hyperedge) has a team performance score associated with it. This team performance score is a real number, equal to the number of points earned by the team while performing one or more tasks within a gaming session. We treat the embedding learned for a given team (hyperedge) as its feature vector which is associated with a real number (team performance). We therefore, perform on regression over all the hyperedges (teams) with team performance as the dependent variable.

Similarly, in **LangNet** dataset each phrase (hyperedge) has a sentiment score associated with it, which again is a real number. Similar to the team dataset above, we treat the embedding learned for a given phrase (hyperedge) as its feature vector which is associated with a real number (sentiment score). We therefore, treat this as a regression task with sentiment score as the dependent variable and perform regression using the feature matrix containing embeddings of all the phrases.

For both the tasks we just described, we perform several evaluation runs. In each run we randomly choose 30% of hyperedges (teams or phrases) as the test set and learn ridge regression parameters using the remaining 70% training hyperedges for each of the 38 different embedding scenarios. Root mean squared error (RMSE) was chosen as the evaluation metric (the lower, the better). RMSE was calculated for each of the 38 scenarios and for each run. Final RMSE score was taken as the average RMSE score across five runs. Ridge regression's hyper-parameter was chosen by 5-fold cross-validation.

For the auto-encoder method (**h2v-auto**) we consider three scenarios: (1) single hidden layer ($L1$) of $d = 128$; (2) two hidden layers ($L1$ & $L2$) with size of $L1 : d = 96$ and of $L2 : d = 32$. We concatenate these embedding to get a single $d = 128$ size embedding; and (3) two hidden layers ($L1$ & $L2$) with size of $L1 : d = 512$ and of $L2 : d = 128$. We use the output of $L2$, which is of dimension $d = 128$, as the embedding. For sampling, we use the distance parameter $h = 2$ for generating the sub-lattice and $\tau = 0.2$ for both datasets. Also, $p = 10$ & $p = 5$ number of hyperedges are sampled (corresponding to each hyperedge) from **EQII** and **LangNet**, respectively.

### 4.3 Results and Discussion

Tables 1 & 2 show the RMSE scores of **t2v** (as compared with baselines) for the tasks of team performance prediction and sentiment score prediction, respectively. These tables contain scores for all the 35 different scenarios: columns represent 7 (6 baselines and the proposed *hypergraph tensor decomposition* (**t2v**)) different models while rows represent combination strategies. The scores for *hypergraph auto-encoder* (**h2v-auto**) are shown in the separate Table 4 as auto-encoder only generates hyperedge embeddings.

**Accuracy & Run-times** As we can observe that for our datasets and for both the regression tasks, almost all the embeddings (baselines and proposed) are performing very similarly in terms of accuracy. However, we can observer in Table 3 that tensor based method (**t2v**) have significant less time than the expensive auto-encoder technique (**h2v-auto**) (Table 4). Among the baselines **e2v-hyp** is the fastest. We also observer that sentence based techniques run faster on text-based **LangNet** dataset as compared to node2vec based methods which are not designed for text data and vice-versa.

**Interpretability** All the matrix or tensor based algebraic techniques: **e2v**, **e2v-hyp** and **t2v**, have a natural graph theoretic interpretation. **e2v** and **e2v-hyp** are both well studied spectral techniques with several eigen-value based interpretations. **t2v** has a hierarchical *hasse diagram* based interpretation. In contrast the sentence and node2vec based techniques can be understood only intuitively in terms of the cost function. Apart from the noise which has a random walk based interpretation, **h2v-auto** is a deep-learning based method which exploits multiple level of non-linearity and is over all a black-box approach.

**Information Loss** To reiterate one of the primary aims of this study is to design methods that retain the hyperedge-level joint information. The proposed tensor-based **t2v** principally capture the joint distribution over various cardinality hyperedges unlike conditional distribution like sentence embedding (**h2v-dm**) (which are more appropriate for sequences). In comparison the other method proposed **h2v-auto**, although are not directly designed to retain hyperedge-level joint distribution, but we hypothesize that the deep layered neural network should output highly informative non-linear representations. Several other baselines only retain pair-wise information. For example, the methods developed using Node2Vec in Section A.2 are based on the skip-gram model, which learns embedding of nodes while maximizing the conditional probability of a node given another node in a context (Eq. 17). Similarly, the spectral methods (Section A.3) are inherently two dimensional as they are based on matrix. Same is the case with skip-gram based sentence embedding (**h2v-dbow**). However, sentence embedding based on the **DM** architecture (**h2v-dm**) maximizes the conditional probability of a word given the previous set of words (*context*) without breaking this context (by concatenating or averaging the embedding of the previous words).

**Leveraging Hypergraph Topology** is another feature that we stress as a desirable property in a method which aims to build hyperedge embedding. Hypergraph topology is an important auxiliary information, if left unused is wastage of resources at hand. We observe that except the sentence-based baselines, both proposed methods as well as other baselines, which are adapted for the hypergraph setting, leverage hypergraph topology using various hypergraph representations. Proposed **h2v-auto** generated noise using the *hasse lattice* and **t2v** directly models the hasse lattice by storing hyperedge level joint distribution using super-symmetric tensors. Node2vec based and spectral baselines use various kinds of matrices (that capture hypergraph topology up to varying degrees) as enlisted in (3-6) in Section 4.2.1.

**Ability to generate hyperedge embeddings directly** is another critical aim as was highlighted in the introduction. We emphasize that none of the baselines are designed to give hyperedge embeddings in a principled manner. However, both tensor based (**t2v** using $\mathcal{A}_{\mathbf{dual}}^{\mathbf{k}}$) as well as auto-encoder (**h2v-auto**) methods proposed in this paper are specifically designed to give hyperedge representations directly.

**Choice of Method** Both tensor based (**t2v** using $\mathcal{A}_{\mathbf{dual}}^{\mathbf{k}}$) as well as auto-encoder (**h2v-auto**) methods proposed in this paper are able to generate embeddings for all hyperedges of various cardinalities present in data. However, the auto-encoder method *cannot* generate node embeddings unlike tensor method (using $\mathcal{A}_{\mathbf{hyp}}^{\mathbf{k}}$). On contrary, in auto-encoder technique we don't need to figure out $\alpha$ and $\beta$ parameters (as discussed in the next subsection 4.4). Another, thing to note is that tensors are inherently capable of retaining joint information because of their higher-order structure. Auto-encoder method implicitly (possibly lossy manner) leverage the higher order information embedded in network in form of node weights ($R(g_i)$) while performing the random walk sampling. Main advantage of auto-encoders is the crisp embedding achieved via multiple levels of non-linearity offered by deep neural networks. Although this non-linearity improves the accuracy of auto-encoder methods very slightly (Table 4) as compared to tensor method (bottom right corner of (Table 1 & 2), the computation cost of tensor is far less than the auto-encoder method (Table 3 & 4). Intuitively, it seems that depending upon the task at hand, in some tasks the retention of joint hyperedge-level information is more important while modeling and data has lesser non-linear structure; or vice versa. But the low computational cost (as reflected in our tasks) makes tensor method definitely more lucrative.

Table 1: RMSE Scores of **(t2v)** compared to baselines for **EQ II** Team Performance Analysis

| | Baselines | | | | | | Hypergraph |
|---|---|---|---|---|---|---|---|
| | Sentence Embed based | | Node2Vec based | | Spectral methods | | Tensor Decomp. |
| **Embed Combination** | **h2v-DM** | **h2v-DBOW** | **h2v-inv** | **h2v-dual** | **e2v** | **e2v-hyp** | **(t2v)** |
| **Node Embed Sum** | 0.79308 | 0.79567 | 0.80418 | 0.79956 | 0.81183 | 0.81405 | 0.81341 |
| **Node Embed Sum + Hyperedge Embed** | 0.79651 | 0.80241 | 0.81362 | 0.80636 | 0.8113 | 0.81652 | 0.81299 |
| **Node Embed Average** | 0.81584 | 0.81733 | 0.82407 | 0.82281 | 0.81234 | 0.81369 | 0.81303 |
| **Node Embed Avg + Hyperedge Embed** | 0.8182 | 0.82077 | 0.83378 | 0.82896 | 0.81223 | 0.81608 | 0.8127 |
| **Only Hyperedge Embed** | 0.81203 | 0.81522 | 0.82189 | 0.81984 | 0.81233 | 0.81608 | 0.81341 |

Table 2: RMSE Scores of **(t2v)** compared to baselines for **LangNet** Sentiment Analysis

| | Baselines | | | | | | Hypergraph |
|---|---|---|---|---|---|---|---|
| | Sentence Embed based | | Node2Vec based | | Spectral methods | | Tensor Decomp. |
| **Embed Combination** | **h2v-DM** | **h2v-DBOW** | **h2v-inv** | **h2v-dual** | **e2v** | **e2v-hyp** | **(t2v)** |
| **Node Embed Sum** | 0.14081 | 0.14029 | N/A | N/A | 0.14633 | 0.14854 | 0.14194 |
| **Node Embed Sum + Hyperedge Embed** | 0.14028 | 0.13883 | N/A | N/A | 0.14627 | 0.14845 | 0.14144 |
| **Node Embed Average** | 0.14245 | 0.14115 | N/A | N/A | 0.14665 | 0.14852 | 0.14381 |
| **Node Embed Avg + Hyperedge Embed** | 0.14178 | 0.14007 | N/A | N/A | 0.14661 | 0.14845 | 0.14333 |
| **Only Hyperedge Embed** | 0.14194 | 0.14147 | N/A | N/A | 0.14744 | 0.14844 | 0.1482 |

Table 3: Average Runtime (seconds) of **(t2v)** compared to baselines across datasets

| | Baselines | | | | | | Hypergraph |
|---|---|---|---|---|---|---|---|
| | Sentence Embed based | | Node2Vec based | | Spectral methods | | Tensor Decomp. |
| **Dataset** | **h2v-DM** | **h2v-DBOW** | **h2v-inv** | **h2v-dual** | **e2v** | **e2v-hyp** | **(t2v)** |
| **EQ2** | 455.84 | 103.47 | 90.05 | 93.41 | 128.03 | 12.01 | 213.37 |
| **LangNet** | 80.61 | 62.31 | 211.97* | 207.86* | 221.46 | 47.12 | 483.81 |

\* these are average time taken for learning vertex embeddings only

Table 4: RMSE Scores & Run-times for **EQ II** (Team Performance) & **LangNet** (Sentiment Analysis) using *Hypergraph Auto-encoder* (**h2v-auto**)

| | EQ II | | | LangNet | | |
|---|---|---|---|---|---|---|
| **Layer Sizes** | **L1:128** | **L1:96/L2:32** | **L1:512/L2:128** | **L1:128** | **L1:96/L2:32** | **L1:512/L2:128** |
| **RMSE** | 0.81104 | 0.81512 | 0.81635 | 0.14568 | 0.14529 | 0.14784 |
| **Run Time** | 52 min | 40 min | 1 hr 20 min | 2 hr 10 min | 3 hr 20 min | 6 hr |

Together with the natural graph theoretic interpretation, tensor methods in comparison to black-box approach of deep learning based auto-encoder, makes them even more viable.

**Highlight** Another distinguishing feature of our work is that we leverage the existing structure (group structure) within the data. Recent attempts along these lines in network literature (Grover & Leskovec, 2016; Tang et al., 2015b; Perozzi et al., 2014) argue that language models have ready-made context in the form of sentences or paragraphs to train the model, which are not available in networks, and therefore, they propose different ways to generate this context. In contrast, we focus on networks where the context is already present, e.g. collaboration networks where collaborative teams are hyperedges, or language hyper-networks where sentences are hyperedges.

## 4.4 SCALABILITY

Both tensor based (**t2v** using $\mathcal{A}^{\mathbf{k}}_{\mathbf{dual}}$) as well as auto-encoder (**h2v-auto**) methods proposed in this paper are able to generate embeddings for *all hyperedges* of various cardinalities present in data.

**Scalability for t2v** The scalability issue in tensor based approach arise on two fronts: (1) the increase in number of hyperedges ($m$) or number of vertices ($n$) and (2) increase in parameter $\alpha$ (maximum cardinality hyperedge considered in the cost function) or $\beta$ (maximum degree vertex considered in the cost function). Also there are two kinds of cost: (1) enumeration cost associated associated while building hypergraph and dual tensors and (2) cost for hypergraph tensor decomposition.

In case of hyperedge embeddings, for each vertex $v$ of degree (i.e. number of hyperedges it is part of) $d(v)$ there are $d(v)!$ entries (permutations of hyperedges incident on $v$) to be enumerated to fill the tensor $\mathcal{A}^{\mathbf{d(v)}}_{\mathbf{dual}}$. This amounts to a worst case enumeration cost of $\sum_{v \in V} d(v)!$. However, $d(v)!$ can grow prohibitively large for vertices with large vertex degree $d(v)$. We therefore, propose to restrict ourselves to vertices of degree $d(v) \leq \alpha$. In fact we propose the following augmented initialization scheme for the dual tensors. For degree $k$ ($\leq \alpha$) vertex, $v_j \in \{g_{q_1}, g_{q_2}, ..., g_{q_k}\}$ and $d(v_j) = k \leq \alpha, \forall j \in \{1, ..., n\}$, we initialize the following elements of $\mathcal{A}^{\mathbf{k}}_{\mathbf{dual}} = (a_{q_1,q_2,...,q_k}) \in \mathbb{R}^{[k,m]}$, same as Equation 29. For degree $k$ ($> \alpha$) vertex, $v_j \in \{g_{q_1}, g_{q_2}, ..., g_{q_k}\}, \forall j \in \{1, ..., n\}$, we initialize elements of $\mathcal{A}^{\alpha}_{\mathbf{dual}}$ as,

$$a_{q_1,q_2,..,q_\alpha} = 1$$
$$a_{q_{(\alpha+1)},q_{(\alpha+2)},.....,q_{2\alpha}} = 1$$
$$.......$$
$$.......$$ \hfill (8)
$$a_{q_{((\lfloor \frac{k}{\alpha} \rfloor - 1)\alpha + 1)}, q_{((\lfloor \frac{k}{\alpha} \rfloor - 1)\alpha + 2)}, ......, q_{((\lfloor \frac{k}{\alpha} \rfloor)\alpha)}} = 1$$

and if $\delta = (k - (\lfloor \frac{k}{\alpha} \rfloor \alpha)) \neq 0$, then we also initialize the following element of $\mathcal{A}^{\delta}_{\mathbf{dual}}$ as:

$$a_{q_{((\lfloor \frac{k}{\alpha} \rfloor)\alpha + 1)}, q_{((\lfloor \frac{k}{\alpha} \rfloor)\alpha + 2)}, ......, q_k} = 1. \hfill (9)$$

The above initialization basically partitions the $k$ hyperedges incident on a vertex into $(\lfloor \frac{k}{\alpha} \rfloor + 1)$ (or $(\lfloor \frac{k}{\alpha} \rfloor)$, if $\alpha$ is a divisor of $k$) partitions. There are several techniques to obtain such a partition, but in this paper we simply sort the incident hyperedges by their cardinality and then sequentially make sets of $\alpha$ elements each (except the last partition which is of size $(k - (\lfloor \frac{k}{\alpha} \rfloor \alpha))$ is not an integer) as shown in Equations( 8- 9). This augmentation shall decrease the enumeration cost to: $\mathbf{\Delta_{dual}} = \sum_{v \in V_a} d(v)! + \sum_{v \in V_b} \left( \lfloor \frac{d(v)}{\alpha} \rfloor \alpha! + (d(v) - (\lfloor \frac{d(v)}{\alpha} \rfloor \alpha))! \right)$, where $V_a = \{v | v \in V, d(v) \leq \alpha\}$ and $V_b = (V - V_a)$. Depending upon the availability of resources (single machine or a distributed computing resources) we can choose the cut-off degree ($\alpha$) accordingly. Notice that although we perform this degree thresholding we still get embedding for all the hyperedges. However, what we sacrifice is the higher order information that a given vertex connects a set of incident hyperedges jointly. But for higher order vertices we argue that as the vertex degree get too high our belief in the inference that hyperedges incident are related diminishes. These high degree vertices can be interpreted as highly popular person participating in a huge number of teams, but should the embeddings of these teams be related to each other is difficult to infer. In text data these high degree nodes correspond to words occurring in a large number of phrases, but this doesn't mean the sentences are related, rather its just that this word is quiet common.

In case of vertex embeddings, for each hyperedge $g_i \in G$, we need to initialize $|g_i|!$ elements of $\mathcal{A}_{\mathbf{hyp}}^{|\mathbf{g_i}|}$, with a worst case enumeration cost of $\sum_{g_i \in G} |g_i|!$. Given this cost can be prohibitive as the hyperedge cardinalities increase, we propose an augmented initialization scheme for hypergraph tensors where we shall restrict ourselves to hyperedges of cardinality $\leq \beta$. For hyperedges $g_i$ with $|g_i| = k \leq \beta$ we initialize $\mathcal{A}_{\mathbf{hyp}}^{\mathbf{k}} = (a_{p_1, p_2, .., p_k}) \in \mathbb{R}^{[k,n]}$ same as Equation 26. For hyperedge with cardinality $|g_i| > \beta$, we initialize the elements of $\mathcal{A}_{\mathbf{hyp}}^{\beta} \in \mathbb{R}^{[\beta,n]}$ as follows:

$$a_{p_1^l, p_2^l, .., p_\beta^l} = \frac{R(g_i)}{\gamma} \tag{10}$$

where $(p_1^l, p_2^l, .., p_\beta^l)$ is the $l$-th permutation among $l = \{1, 2, ..., \gamma\}$, where $\gamma = \binom{|g_i|}{\beta}$. Enumeration cost now decreases to: $\boldsymbol{\Delta}_{\mathbf{hyp}} = \sum_{g_i \in G_a} |g_i|! + \sum_{g_i \in G_b} \binom{|g_i|}{\beta} \beta!$, where $G_a = \{g_i | g_i \in G, |g_i| = \beta\}$ and $G_b = (G - G_a)$. The choice of cut-off ($\beta$) is therefore, dictated by the computation resources available. Notice that although we perform this cardinality thresholding we still get embedding for all the vertices. However, what we sacrifice is the higher order information that a given hyperedge connects a set of incident vertices jointly. Now, as the $k$ increases, the number of $k$ cardinality hyperedge decrease as well as the $\mathcal{A}_{\mathbf{hyp}}^{\mathbf{k}}$ tensor grows exponentially making it increasingly sparse and less informative (in information theoretic sense). Therefore, restricting to $\beta$ cardinality hyperedges is a trade-off between enumeration cost and hyperedge level joint information loss. We observer that enumeration involved in both hyperedge and vertex embeddings, Equations( 8- 10), are either *vertex or hyperedge centric computations* and can be performed in a scalable manner using the generic hyperedge-centric distributed computation libraries like MESH (Heintz & Chandra, 2015; Heintz et al., 2017) and HyperX (Huang et al., 2015).

Now we consider the tensor decomposition complexity and its scalability. For hyperedge embeddings, the per iteration complexity of the algorithm is $O(\sum_{\delta=1}^{\alpha} \mathrm{nnz}(\mathcal{A}_{\mathbf{dual}}^{\delta}))$. Notice, $\left(\sum_{\delta=1}^{\alpha} \mathrm{nnz}(\mathcal{A}_{\mathbf{dual}}^{\delta})\right) \leq \boldsymbol{\Delta}_{\mathbf{dual}}$. Similarly, in case of vertex embeddings the per iteration complexity of the algorithm is $O(\sum_{k=c_{min}}^{\beta} \mathrm{nnz}(\mathcal{A}_{\mathbf{hyp}}^{\mathbf{k}}))$. Notice, $\left(\sum_{k=c_{min}}^{\beta} \mathrm{nnz}(\mathcal{A}_{\mathbf{hyp}}^{\mathbf{k}})\right) \leq \boldsymbol{\Delta}_{\mathbf{hyp}}$. Hypergraph Tensor Decomposition basically involves learning vertex or hyperedge embeddings, i.e. parameters which are vertex or hyperedge centric. We can therefore, convert the tensor decompositions into equivalent hyperedge or vertex centric message passing algorithm and use the generic hyperedge or vertex centric distributed computation libraries like MESH or HyperX, as mentioned previously. Although, we highlight the possible directions for scalability, but we propose them as a future work.

**Scalability without partitioning for t2v** As discussed previously, as we are performing hyperedge cardinality as well as vertex degree thresholding and for that reason we developed augmented enumeration schemes in Equation ( 8- 10). The partitioning involved in this augmentation ensures that all the hyperedges and all the vertices are linked by some vertex of degree $\leq \alpha$ and some hyperedge of cardinality $\leq \beta$. If this augmentation was not performed, and we simply use the enumeration of Equations ( 26- 29), it was possible for example, that some hyperedge which has no vertex whose degree is $\leq \alpha$, and therefore, would be left out and no embedding would be learned for this hyperedge (because of cold start in tensor decomposition). Rather than performing the augmentation we can use semi-supervised learning to learn the embeddings for *critical hyperedges* (hyperedges with all vertices with degree $> \alpha$) using the embeddings of the *non-critical hyperedges* ((hyperedges with at least one vertex with degree $\leq \alpha$)). As all the hyperedges (critical or not) are linked using the hypergraph topology, we can perform semi-supervised learning using topology based regularization while performing tensor decomposition. This can be done using a graph regularization term $((_k\mathbf{U}^{(\mathbf{j})^T})\mathbf{L}(_k\mathbf{U}^{(\mathbf{j})}))$ in our tensor decomposition objective function 5. Here, $\mathbf{L}$ is the graph laplacian of our choice. For example we can use $\mathbf{L}_{\mathbf{hyp}}$ or $\mathbf{L}_{\mathbf{graph}}$ when we perform tensor decomposition for vertex embeddings and use $\mathbf{L}_{\mathbf{inv}}$ or $\mathbf{L}_{\mathbf{dual}}$ for hyperedge embeddings. Appendix A.3 mentions the details about these laplacians.

**Scalability for h2v-auto** In case of auto-encoder method the main scalability challenge lies in generating the noisy hyperedges. For a given hyperedge the intermediate sub-lattice from which $p$ samples are drawn is of size $O(n^h)$, where $h$ is max distance from the hyperedge considered. Once this sampling is performed we have total $mp$ hyperedges as input for the auto-encoder, which is linear in the number of hyperedge as $p$ will be a constant. The main challenge therefore, is to generate the noisy samples where we have to tackle exponential size sub-lattice per hyperedge. Again,

as this sampling is done for each hyperedge separately (hyperedge-centric), we can therefore, use distributed computing for hypergraphs using MESH and HyperX, as mentioned before.

**Issues with aggregating vertex embedding** Furthermore, we also observe that even simple element-wise summation or averaging of node embeddings for the nodes (in a given hyperedge) also perform comparably when compared to hyperedge embedding alone. From this we can infer that depending upon the dataset, if we have less hyperedges and more nodes, than we would rather prefer to simply learn the hyperedge embedding directly rather than learning node embeddings and then performing aggregating operation over them. Aggregation might turn out be costly specially if average hyperedge size is large and the choice of aggregation function is an issue. Therefore, learning hyperedge embeddings directly seems to be escape the problem of choosing the aggregation function all together.

**Scalability issues for baselines** Another thing we notice, is that in case of **LangNet** dataset, that node2vec based **h2v-inv** and **h2v-dual** methods are simply unable to run (see N/A in Table 2). It seems that in **LangNet** (and possibly in text data in general) the hypergraph dual, $A_{dual}$, which contain phrase to phrase edges turns out be containing significantly more edges than the number of node to node edges in $A_{hyp}$. Therefore, performing context generation (using node2vec) over $A_{inv}$ and $A_{dual}$ graphs turns out be very costly and we were unable to get hyperedge embeddings for **h2v-inv** and **h2v-dual** methods. Note, we however do get vertex embedding (as mentioned with a * mark below Table 3). But as you can see that even the time time taken for vertex embedding is similar or more than the total time taken by spectral methods for learning both hyperedge as well as vertex embedding, together. This indicates the robustness of the spectral methods, specially the **e2v-hyp** for different kinds of hypergraph data and edge densities (sparsity).

# 5 RELATED WORKS

**Hypergraphs** were studied rigorously by Berge (1976; 1984) as a generalization of graphs and *directed* hypergraphs have been introduced by Bretto (2013). Hypergraph were argued for the first time as a model to naturally capture higher-order relationships between sets of objects across variety of domains by Estrada & Rodriguez-Velazquez (2005). Hypergraphs have been used to model complex networks in different fields including biology (Klamt et al., 2009), databases (Fagin, 1983) and data mining (Han et al., 1997; Zhou et al., 2007). Within machine learning, algorithms guided by the structure of hypergraph were introduced by Zhou et al. (2006) and have found applications in a variety of domains (Tian et al., 2009; Gao et al., 2013; Li & Li, 2013; Sharma et al., 2015; 2017). Simplicial complex (Munkres, 1984) based view of hypergraph using *hasse lattice* (Skiena, 1990) within machine learning has recently been proposed by Sharma et al. (2017).

**Representation learning (RL)** (Bengio et al., 2013) focuses on learning features (geometry) of the data (topology). Machine learning algorithms make use of these features for prediction. Traditionally these features are readily available within the data-set or are engineered manually. However, this is often a tedious and human labor-intensive process. RL addresses this issue by learning features automatically in a task-dependent supervised or a task-independent unsupervised manner.

*Node Representations in Graph:* Traditionally unsupervised node embedding learning has been dong using latent models like matrix factorization (Ahmed et al., 2013) or by community detection (Tang & Liu, 2011) based techniques for networks (Roweis & Saul, 2000; Belkin & Niyogi, 2001; Tenenbaum et al., 2000; Cox & Cox, 2000). In each case there is a vector of features learned for a node, each of whose entries reflects node's association with some latent dimension or a network community. More recently, there has been a revived interest in graph embedding in form of *context* oriented techniques (Perozzi et al., 2014; Tang et al., 2015b; Grover & Leskovec, 2016). These techniques are inspired by recent unsupervised RL methods in NLP (Mikolov et al., 2013b; Le & Mikolov, 2014) where word embeddings are learned that are similar to words in a given neighborhood or *context*. These techniques differ in the manner they generate this context as well as in the objective which they optimize. Also there are supervised algorithms learn embeddings which are optimal for the specific task at hand. This results in high accuracy but incurs significant computational cost for training. Recently, several supervised learning algorithms have been proposed for network analysis (Tian et al., 2014; Xiaoyi et al., 2013) and for text networks in a semi-supervised setting (Tang et al., 2015a). Finally, we refer readers to a very recent and comprehensive survey on graph embedding methods by Cai et al. (2017).

*Node Representations in Hypergraph:* Learning embeddings for nodes within a hypergraph while incorporating the hypergraph topology using proxy graphs is introduced by Zhou et al. (2006). Using graph proxy destroys the hypredge-level joint information and thus, incur loss of information. Also, Agarwal et al. (2006) squarely criticize that such representations can be learned by constructing graphs, which are proxies for the hypergraph structure.

*Set Representations:* RL for sets using neural networks has been proposed recently (Vinyals et al., 2016), where a memory network is used to compose features sequentially but in an order invariant manner. In their very recent paper, Rezatofighi et al. (2016) have tried to answer this set ordering issue by the use of *random set theory*. However, they do not consider embedding but focus on learning set-level probabilities. More importantly, both of these works, do not consider the hypergraph structure of overlapping sets which is the main focus of this paper.

**Tensors** For comprehensive view of tensors, tensor decomposition as well as applications we refer to the survey by Kolda & Bader (2009). The connection between $k$-way tensor and k-uniform hypergraph eigen values was established by Qi (2005). The use of $k$-way symmetric tensor and their non-negative decomposition for uniform hypergraph partitioning was first introduced by Shashua et al. (2006). But they are again restricted to uniform hypergraphs.

## 6 CONCLUSION

In this paper we have proposed two methods to generate higher-order representations for both hyperedges (representing sets of nodes) and hypergraph nodes (that also take into account the hypergraph structure). First, is an auto-encoder based deep-learning method and second, is a tensor-based algebraic method. Both learning models are unique in the manner they leverage the existing structure present in network data as context. While introducing a new idea of a dual tensors corresponding to the hypergraph dual, we develop a novel approach of using factors from joint decomposition of $k$-way tensors corresponding to $k$-uniform sub-hypergraphs, as generic node & hyperedge representations. We show that that both methods perform comparably with several other baselines in terms of accuracy. We also observe that the proposed tensor based methods are more efficient and also have a natural hypergraph theoretic interpretation; unlike deep learning based black-box approach. We therefore, argue that we have proposed more general methods which are principally suited for hypergraphs (and therefore also for graphs) while maintaining accuracy and efficiency.

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

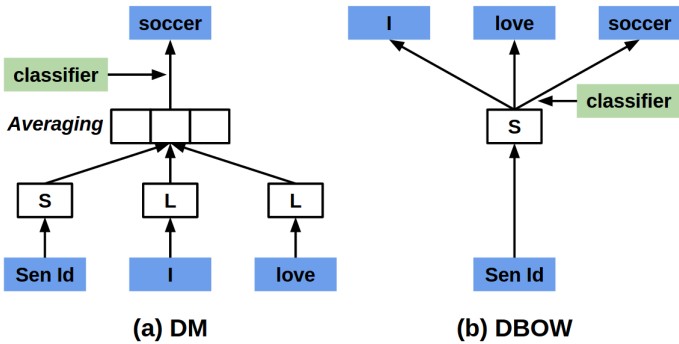

Figure 4: Distributed memory (DM) and Distributed bag of words (DBOW) versions of Sen2Vec for a sentence "I love soccer".

# A APPENDIX

## A.1 HYPEREDGE2VEC USING SENTENCE EMBEDDINGS

As mentioned before, the most commonly used model for studying complex interactions in networks is graphs, where each edge represents a dyadic interaction between nodes Strogatz (2001). Therefore, even if the original interaction is not dyadic like a set of researchers (3 or more) collaborating on a publication, we shall break down this joint interaction into dyadic interactions. Even though a large number of complex network data naturally occurs as hypergraphs Estrada & Rodriguez-Velazquez (2005), the popularity of "think like dyadic edges" and not like sets seems to us as a hindrance, specifically when the end aim is to model the joint distribution at the level of sets. In this paper, as it is obvious, we go by "think like set" paradigm, specifically, when the data is naturally occurring as network of sets (i.e., a hypergraph). Therefore, if we think of hyperedge as a set, a natural question arises is that are there any existing techniques that learn embeddings at set level? We, henceforth, explore the most popular discrete data on which representation learning has been applied is text data.

Recently, Le & Mikolov (2014) proposed two representation learning methods for sentences as shown in Figure 4: a) a distributed memory (**DM**) model, and b) a distributed bag of words (**DBOW**) model. In the DM model, every sentence in the dataset $G$ is represented by a $d$ dimensional vector in a shared lookup matrix $S \in \mathbb{R}^{|G| \times d}$. Similarly, every word in the vocabulary $\Omega$ is represented by a $d$ dimensional vector in another shared lookup matrix $L \in \mathbb{R}^{|\Omega| \times d}$. Given an input sentence $\mathbf{v} = (v_1, v_2 \cdots v_m)$, the corresponding sentence vector from $S$ and the word vectors from $L$ are averaged to predict the next word in a context. More formally, let $\phi$ denote the mapping from sentence and word ids to their respective vectors in $S$ and $L$, the DM model minimizes the following objective:

$$J(\phi) = -\sum_{t=k}^{m-k} \log P(v_t | \mathbf{v}; v_{t-k+1}, \cdots, v_{t-1}) \tag{11}$$

$$= -\sum_{t=k}^{m-k} \log \frac{\exp(\omega(v_t)^T \mathbf{z})}{\sum_i \exp(\omega(v_i)^T \mathbf{z})} \tag{12}$$

where $\mathbf{z}$ is the average of $\phi(\mathbf{v}), \phi(v_{t-k+1}), \cdots, \phi(v_{t-1})$ *input* vectors, and $\omega(v_t)$ is the *output* vector representation of word $v_t \in \Omega$. The sentence vector $\phi(\mathbf{v})$ is shared across all (sliding window) contexts extracted from the same sentence, thus acts as a distributed memory.

Instead of predicting the next word in the context, the DBOW model predicts the words in the context independently given the sentence id as input. More formally, DBOW minimizes the following objective:

$$J(\phi) \quad = \quad -\sum_{t=k}^{m-k} \sum_{j=t-k+1}^{t} \log P(v_j|\mathbf{v}) \tag{13}$$

$$= \quad -\sum_{t=k}^{m-k} \sum_{j=t-k+1}^{t} \log \frac{\exp(\omega(v_j)^T \phi(\mathbf{v}))}{\sum_i \exp(\omega(v_i)^T \phi(\mathbf{v}))} \tag{14}$$

Both the methods take sentences as input and return embeddings for words as well as the sentence. To apply the sentence embedding models directly to hyperedges, as our first method we generate a proxy sentence for each hyperedge by leveraging the contextual information in the hyperedge.

A proxy sentence $\mathbf{v}_i$ is formed for each hyperedge $g_i \in G$ as a sequence made by concatenating all the permutations of the nodes (as words) in the hyperedge and further repeating this sequence as many time this hyperedge occurred. For example, for a three node hyperedge $g_i = \{1, 4, 7\}$ which has occurred two times we make the following sentence $\mathbf{v}_i$: {1,4,7,1,7,4,7,1,4,7,4,1,4,1,7,4,7,1,1,4,7,1,7,4,7,1,4,7,4,1,
4,1,7,4,7,1} of length 12 (6 permutations times 2 occurrence). In this scheme we have permutations and repetitions. We take all the permutations hoping that it should fool the sequence based model to get a sequence independent embedding. On the other hand, repetition of the permutations acts similarly to observing the same sequence of words several times in the text corpus. Of course one can think of other alternative schemes, but the point we are trying to make is that naturally observed hyperedges captures important contextual information that can be leveraged to achieve better representations. We refer to the node-hyperedge embedding pairs resulting from **DM**, **DBOW** models as **h2v-dm** and **h2v-dbow**, respectively.

### A.2 HYPEREDGE2VEC USING NODE2VEC

Grover & Leskovec (2016) propose a representation method for nodes in a graph called *node2vec*, which uses the skip-gram model Mikolov et al. (2013b) with the intuition that neighborhood nodes within a graph should have similar representations. The objective of the skip-gram model for graphs can be defined as:

$$J(\phi) \quad = \quad -\sum_{v \in V} \log P(N(v)|\phi(v)) \tag{15}$$

$$= \quad -\sum_{v \in V} \sum_{n_i \in N(v)} \log P(n_i|\phi(v)) \tag{16}$$

$$= \quad -\sum_{v \in V} \sum_{n_i \in N(v)} \log \frac{\exp(\omega(n_i)^T \phi(v))}{\sum_{x \in V} \exp(\omega(x)^T \phi(v))} \tag{17}$$

where as before, $\phi$ and $\omega$ denote the *input* and the *output* vector representations of the nodes. The neighboring nodes $N(v)$ form the *context* for node $v$. node2vec uses a biased random walk which adaptively combines breadth first search (BFS) and depth first search (DFS) to find the neighborhood of a node. The walk attempts to capture two properties of a graph often used for prediction tasks in networks: (i) homophily and (ii) structural equivalence. According to homophily, nodes in the same group or community should have similar representations. Structural equivalence suggests that nodes with similar structural roles (hub, bridge) should have similar representations. In a real-world network, nodes exhibit mixed properties.

Our aim is to find both node and hyperedge level embeddings by taking into account the hypergraph structure. For the former, we leverage the adjacency matrix associated with a hypergraph Zhou et al. (2006), which is defined as:

$$\mathbf{A_{hyp}} = \mathbf{H}^T \mathbf{W_e} \mathbf{H} - \mathbf{D_v} \tag{18}$$

where $\mathbf{W_e}$ is a diagonal matrix containing the weights of each hyperedge and $\mathbf{D_v}$ is a diagonal matrix containing the degree of each vertex. We take the weight of a hyperedge as its occurrence number, i.e. $\mathbf{W_e}(i, i) = R(g_i), \forall \{i \in \{1, ..., n\}\}$. The adjacency matrix ($\mathbf{A_{hyp}} \in \mathbb{R}^{|V| \times |V|}$) associates a weight between a pair of nodes while taking into account the weights of all the hyperedges that encompass a pair of nodes. The weighted graph associated with $\mathbf{A_{hyp}}$ in some sense serves

as a proxy to the actual hypergraph structure. We can provide $\mathbf{A_{hyp}}$ as input to node2vec and hypothesize that the random walk over this proxy hypergraph should allow the skip-gram model to learn more meaningful node level embeddings which can be combined to construct hyperedge level embeddings. We refer to these node level embeddings as **N2V-hyp**.

We still have not met our second objective to learn hyperedge embeddings directly. We again wish to leverage node2vec for our purpose. However, node2vec only works for graphs by performing random walk over nodes. Therefore, we ask ourselves the question, that can we treat hyperedges as nodes? There can be other ways of doing so, but here we suggest two techniques. In the first technique, we simply consider the *hypergraph dual* Berge (1984), whose incidence matrix is $\mathbf{H_{dual}} = \mathbf{H}^T$. The adjacency matrix associated with the hypergraph dual is:

$$\mathbf{A_{dual}} = \mathbf{H_{dual}}^T \mathbf{W_v} \mathbf{H_{dual}} - \mathbf{D_e} = \mathbf{H} \mathbf{W_v} \mathbf{H}^T - \mathbf{D_e} \qquad (19)$$

where $\mathbf{W_v}$ is a diagonal matrix containing the weights of each node and $\mathbf{D_e}$ is a diagonal matrix containing the degree of each hyperedge. We assume no weights on the nodes and take $\mathbf{W_v} = \mathbf{I}$. The matrix $\mathbf{A_{dual}} \in \mathbb{R}^{|G| \times |G|}$ represents another hypergraph, but the roles of nodes and hyperedges have now switched. For example, in case of words and sentences, the hyperedges were sentences, but in the hypergraph dual the words become hyperedges and the nodes within a word's hyperedge represent all the sentences in which the word has appeared. We again give $\mathbf{A_{dual}}$ (i.e., a graph proxy for the dual hypergraph) as input to node2vec, but this time we get output as the embeddings for the hyperedges associated with the nodes in dual. We refer to these hyperedge embeddings as **h2v-dual**.

In the second technique, we consider the following adjacency matrix $\mathbf{A_{inv}} \in \mathbb{R}^{|G| \times |G|}$:

$$\mathbf{A_{inv}} = \mathbf{H} \mathbf{H}^T \qquad (20)$$

associated with what we refer to as the *inverted hypergraph*. This inverted hypergraph is a graph (unlike the dual which is a hypergraph) and there is an edge between two nodes if the hyperedges corresponding to the nodes in the original hypergraph have nodes in common. Weight of this edge is the number of common nodes. We again give $\mathbf{A_{inv}}$ as input to node2vec to get embeddings for the hyperedges associated with the nodes in inverted hypergraph. We refer to these hyperedge embeddings as **h2v-inv**.

### A.3 HYPEREDGE2VEC USING SPECTRAL EMBEDDINGS

These set of methods extract embeddings as the eigenvectors associated with Laplacian matrices corresponding to the various adjacency matrices discussed in the previous section. We consider the following graph Laplacians:

$$\mathbf{L_{graph}} = \mathbf{I} - \mathbf{D_v}^{-1/2} \mathbf{A_{graph}} \mathbf{D_v}^{-1/2} \qquad (21)$$

where $\mathbf{A_{graph}} = \mathbf{H}^T \mathbf{W_e} \mathbf{H}$ is the weighted graph associated with the graph corresponding to adjacency matrix $\mathbf{A}$,

$$\mathbf{L_{hyp}} \;=\; \mathbf{I} - \mathbf{D_v}^{-1/2} \mathbf{A_{hyp}} \mathbf{D_v}^{-1/2} \qquad (22)$$
$$\mathbf{L_{inv}} \;=\; \mathbf{I} - \mathbf{D_e}^{-1/2} \mathbf{A_{inv}} \mathbf{D_e}^{-1/2} \qquad (23)$$
$$\mathbf{L_{dual}} \;=\; \mathbf{I} - \mathbf{D_e}^{-1/2} \mathbf{A_{dual}} \mathbf{D_e}^{-1/2} \qquad (24)$$

We get the $d$ eigenvectors associated with the smallest $d$ eigenvalues of the above graph Laplacians as the embeddings. We get vertex embeddings using Eq. 22 and hyperedge embedding using Eq. 24, and we refer to them together as **e2v-hyp**. Similarly, we get another pair of vertex embeddings (using Eq. 21) and hyperedge embedding (using Eq. 24). We refer to the later pair as **e2v**.

## B APPENDIX

### B.1 TENSOR INITIALIZATION SCHEMES

We can also have other initialization for *hypergraph tensor*, $\mathcal{A}_{\mathbf{hyp}}^{\mathbf{k}} = (a_{p_1, p_2, .., p_k}) \in \mathbb{R}^{[k, n]}$, as follows:

$$a_{p_1, p_2, .., p_k} = \frac{R(g_i)}{k!} \qquad (25)$$

where $\{v_{p_1}, v_{p_2}, ..., v_{p_k}\} \in g_i$ and $|g_i| = k, \forall i \in \{1, ..., m\}$. This initialization normalizes the effect of occurrence counts of hyperedge $g_i$ across all of its vertex permutations. It can be interpreted as each vertex permutation marginally contributes to the total occurrence count of the hyperedge. Now if a hyperedge is large intuitively they are less informative about a joint relationship between the vertices part of it. For example the members of a large social group are less likely to be related as compared to members of a small research collaboration who have published several papers together. To incorporate this large hyperedge issue, we normalize the occurrence count with the hyperedge cardinality, as follows:

$$a_{p_1, p_2, .., p_k} = \frac{R(g_i)}{k! \times k} \tag{26}$$

In a similar manner we can also have different initialization schemes for *dual tensor*, $\mathcal{A}_{\mathbf{dual}}^{\mathbf{k}} = (a_{q_1, q_2, .., q_k}) \in \mathbb{R}^{[k,m]}$, as follows:

$$a_{q_1, q_2, .., q_k} = \frac{1}{k!} \tag{27}$$

where $\{g_{q_1}, g_{q_2}, ..., g_{q_k}\} \ni v_j$ and $d(v_j) = k, \forall j \in \{1, ..., n\}$. This initialization indicates that vertex $v_j$ connects various permutations of the hyperedges $\{g_{q_1}, g_{q_2}, ..., g_{q_k}\}$, each contributing fractionally. We can also have a weighted initialization like:

$$a_{q_1, q_2, .., q_k} = \frac{\sum_{s=1}^{k} w_1(g_{q_s})}{k!} \tag{28}$$

If the weight $w_1(g_{q_s}) = R(g_{q_s})$, then the numerator $\sum_{s=1}^{k} w(g_{q_s})$ would be high if the vertex $v_j$ participated in its neighboring hyperedges at several occurrences, therefore, is a strongly informative if we were to infer that the hyperedges $\{g_{q_1}, g_{q_2}, ..., g_{q_k}\}$ are related jointly to each other. For example, if an individual has participated a lot in a bunch of social groups then its reasonable to assume that these groups are related. Same argument goes for a common word in several sentence. However, if a word or a person is part of large sentences or big groups then this argument might not be that strong. Reason being, for example a student can be part of a big university and a big company where he works as a part time employee. This does not make the company and the university related. Another issue is that if a word or a person is very common or popular i.e. the vertex $v_j$ has high degree $k(= d(v_j))$ then its doesn't mean the sentences or groups they are part of are related. Its just that they are common word or popular person that happen to be in a lot of sentence or groups. We can mitigate the above two issues by the following augmented initialization:

$$a_{q_1, q_2, .., q_k} = \frac{\sum_{s=1}^{k} w_2(g_{q_s})}{k! \times k} \tag{29}$$

where $w_2(g_{q_s}) = \frac{R(g_{q_s})}{|g_{q_s}|}$ is the occurrence counts normalized by the hyperedge size.

## B.2 Incident Hyperedge Partitioning Schemes for High Degree Vertices

Here we briefly discuss various partitioning schemes capturing different intuitive objectives. As mentioned in the Section 4.4 we wish to limit ourselves to vertices of degree $d(v) \leq \alpha$. One straight forward method is to simply ignore the vertices with $d(v) > \alpha$. However, this would result in loss of the information. In the sense that each vertex connects a set of incident hyperedges and this connection implies that these set of hyperedges are related to each other and that too in a joint fashion with this vertex acting as a common bridge. Therefore, we don't wish to ignore vertices with $d(v) > \alpha$ but rather partition the $d(v)$ incident hyperedges into sets that have size $\leq \alpha$. Here we enlist a few possible ways:

1. **Linear Partitioning** In this we order the hyperedges in a certain order and then equally partition them into $\lfloor \frac{d(v)}{\alpha} \rfloor$ number of $\alpha$ size partitions (if $\alpha$ is a divisor of $d(v)$), else there shall be one more partition of size $\{d(v) - (\lfloor \frac{d(v)}{\alpha} \rfloor)\}$. There can be various ways to order:

    (a) Order by hyperedge cardinality ($|g_i|$)
    (b) Order by hyperedge occurrence count ($R(g_i)$)
    (c) Order by hyperedge occurrence count normalized by cardinality ($\frac{R(g_i)}{|g_i|}$)

(d) Segregate the *critical hyperedges* for vertex $v$ (i.e. the set $\{g_i | g_i \in G, v \in g_i, d(x) > \alpha, \forall x \in g_i\}$) followed by the remaining hyperedges.

2. **Subset based partitioning** In this we take subsets of the $d(v)$ hyperedges incident on vertex $v$. This can be done in couple of different ways:

   (a) Take all the $\binom{d(v)}{\alpha}$ subsets of $d(v)$ hyperedges and initialize the various permutation for each subset in the tensor $\mathcal{A}^{\alpha}_{\mathbf{dual}}$.

   (b) Take all the $\binom{d(v)}{\alpha}$, $\binom{d(v)}{\alpha-1}$,...., $\binom{d(v)}{2}$ subsets of $d(v)$ hyperedges and initialize the various permutation for each subset in the tensors $\mathcal{A}^{\alpha}_{\mathbf{dual}}$,$\mathcal{A}^{\alpha-1}_{\mathbf{dual}}$,...,$\mathcal{A}^{\mathbf{2}}_{\mathbf{dual}}$, respectively.

3. **Sampling subsets** In this we do not exhaustively use the entire set of $d(v)$ hyperedges but rather sample subsets of size $\alpha$ from the $d(v)$ hyperedges. This sampling is performed in proportion to:

   (a) Hyperedge cardinality ($|g_i|$)

   (b) Hyperedge occurrence count ($R(g_i)$)

   (c) Hyperedge occurrence count normalized by cardinality ($\frac{R(g_i)}{|g_i|}$)

   Further, we can also perform the above sampling of subsets from only among the critical hyperedges, or from only among the non-critical hyperedges or in a mixed manner.

