# OpenReview forum: "Hyperedge2vec: Distributed Representations for Hyperedges"
_ICLR.cc/2018/Conference — Reject_

### Official Review · AnonReviewer1 · 2017-11-27
**Computing node embeddings and hypernode embeddings for hypergraphs**

**Rating:** 5
**Confidence:** 3

**Review:**

The paper studies different methods for defining hypergraph embeddings, i.e. defining vectorial representations of the set of hyperedges of a given hypergraph. It should be noted that the framework does not allow to compute a vectorial representation of a set of nodes not already given as an hyperedge. A set of methods is presented : the first one is based on an auto-encoder technique ; the second one is based on tensor decomposition ; the third one derives from sentence embedding methods. The fourth one extends over node embedding techniques and the last one use spectral methods. The two first methods use plainly the set structure of hyperedges. Experimental results are provided on semi-supervised regression tasks. They show very similar performance for all methods and variants. Also run-times are compared and the results are expected. In conclusion, the paper gives an overview of methods for computing hypernode embeddings. This is interesting in its own. Nevertheless, as the target problem on hypergraphs is left unspecified, it is difficult to infer conclusions from the study. Therefore, I am not convinced that the paper should be published in ICLR'18.

* typos
* Recent surveys on graph embeddings have been published in 2017 and should be cited as "A comprehensive survey of graph embedding ..." by Cai et al
* Preliminaries. The occurrence number R(g_i) are not modeled in the hypergraphs. A graph N_a is defined but not used in the paper.
* Section 3.1. the procedure for sampling hyperedges in the lattice shoud be given. At least, you should explain how it is made efficient when the number of nodes is large.
* Section 3.2. The method seems to be restricted to cases where the cardinality of hyperedges can take a small number of values. This is discussed in Section 3.6 but the discussion is not convincing enough.
* Section 3.3 The term Sen2vec is not common knowledge
* Section 3.3 The length of the sentences depends on the number of permutations of $k$ elements. How can you deal with large k ?
* Section 3.4 and Section 3.5. The methods proposed in these two sections should be related with previous works on hypergraph kernels. I.e. there should be mentions on the clique expansion and star expansion of hypergraphs. This leads to the question why graph embeddings methods on these expansions have not be considered in the paper.
* Section 4.1. Only hyperedeges of cardinality in [2,6] are considered. This seems a rather strong limitation and this hypothesis does not seem pertinent in many applications.
* Section 4. For online multi-player games, hypernode embeddings only allow to evaluate existing teams, i.e. already existing as hyperedges in the input hypergraph. One of the most important problem for multi-player games is team making where team evaluation should be made for all possible teams.
* Section 5. Seems redundant with the Introduction.

---

> ### Author Response · Authors · 2018-01-05
> **Thanks for your valuable comments, some of your concerns were due lack of clear baselines which we have addressed**
>
> Dear Reviewer,
>
> Please find our replies in-line below.
>
> * typos
> * Recent surveys on graph embeddings have been published in 2017 and should be cited as "A comprehensive survey of graph embedding ..." by Cai et al.
>
> Ø  This we have taken care
>
> * Preliminaries. The occurrence number R(g_i) are not modeled in the hypergraphs. A graph N_a is defined but not used in the paper.
>
> Ø  We have used it in our methods.
>
> * Section 3.1. the procedure for sampling hyperedges in the lattice should be given. At least, you should explain how it is made efficient when the number of nodes is large.
>
> Ø  This we have taken care of by explaining in further detail the procedure.
>
> * Section 3.2. The method seems to be restricted to cases where the cardinality of hyperedges can take a small number of values. This is discussed in Section 3.6 but the discussion is not convincing enough.
>
> Ø  Exact prediction of a “large” set is rarely found in real world applications. If we need to go beyond size six we can always employ distributed hypergraph computation frameworks like MESH [2] or HyperX [3].
>
> * Section 3.3 The term Sen2vec is not common knowledge
> * Section 3.3 The length of the sentences depends on the number of permutations of $k$ elements. How can you deal with large k ?
>
> Ø This is a baseline method as we have clarified. And therefore, this is an inherent limitation of the baseline which we have have adapted for comparison purposes.  Although, if we still need to use this baseline with large ‘k’ we can again use distributed hypergraph computation frameworks like MESH [2] or HyperX [3] or in general any other distributed computation for scalable enumeration.
>
> * Section 3.4 and Section 3.5. The methods proposed in these two sections should be related with previous works on hypergraph kernels. I.e. there should be mentioned on the clique expansion and star expansion of hypergraphs. This leads to the question why graph embeddings methods on these expansions have not be considered in the paper.
>
> Ø We have considered the most popular normalized hypergraph laplacian, which has been extensively used in various application domains and is considered state of the art. Not just clique/star expansion but there are number of other hypergraph laplacians that can be employed. The paper by Agarwal et. al. [1] lists several such laplacians, and infact shows that number of them are actually equivalent. Specially the clique / star expansion one are shown to be equivalent with the normalized laplacian we have employed in our work. We therefore, leave this exploration of various such hypergraph laplacians which actually work on a proxy graph as something for future work.
>
> * Section 4.1. Only hyperedges of cardinality in [2,6] are considered. This seems a rather strong limitation and this hypothesis does not seem pertinent in many applications.
>
> Ø  Our algorithms in general works for any given cardinality range [c_min,c_max]. In the datasets used we found that a large portion of the hyperedges were found in the range [2,6]. Therefore, for our experimentation purpose this was a suitable choice.  If we need to go beyond size six or any larger c_max, we can always go distributed hypergraph computation frameworks like MESH [2] or HyperX [3].
>
> * Section 4. For online multi-player games, hypernode embeddings only allow to evaluate existing teams, i.e. already existing as hyperedges in the input hypergraph. One of the most important problem for multi-player games is team making where team evaluation should be made for all possible teams.
>
> Ø We think that team formation is a separate problem in its own right. Team performance is one of the problem we have chosen to illustrate the use of hyperedge embedding as a social science application.
>
> * Section 5. Seems redundant with the Introduction.
>
> Ø  We have taken care of this.
>
> References:
>
> [1] Agarwal, Sameer, Kristin Branson, and Serge Belongie. "Higher order learning with graphs." In Proceedings of the 23rd international conference on Machine learning, pp. 17-24. ACM, 2006.
>
> [2] Enabling Scalable Social Group Analytics via Hypergraph Analysis Systems. Benjamin Heintz and Abhishek Chandra. In the USENIX Workshop on Hot Topics in Cloud Computing (HotCloud). Santa Clara, CA. July, 2015.
>
> [3] J. Huang, R. Zhang, and J. X. Yu, “Scalable hypergraph learning and processing,” in Proc. of ICDM, Nov 2015, pp. 775–780.

---

> > ### Comment · AnonReviewer1 · 2018-01-09
> > **Some comments on the rebuttal**
> >
> > thanks for the rebuttal and modifications on the submitted version. Experimental results do not help to choose between the different methods. Large hyperedges exist in real applications for social networks.

---

> > > ### Author Response · Authors · 2018-01-09
> > > **Made further changes in the experiments for scalability and choice of methods**
> > >
> > > Dear Reviewer,
> > >
> > > Thanks a lot for your response.
> > >
> > > We have submitted a newer version of the paper. In section 4.3 we have made further changes and tried to answer the scalability and choice of methods, aspect more specifically.
> > >
> > > We hope that you find our modifications convincing enough.
> > > Please do let us know.
> > >
> > > Sincere Thanks.

---

### Official Review · AnonReviewer3 · 2017-11-27
**interesting, but the presentation needs to be improved**

**Rating:** 5
**Confidence:** 3

**Review:**

This paper studies the problem of representation learning in hyperedges.  The author claims their novelty for using several different models to build hyperedge representations.  To generate representations for hyperedge, this paper proposes to use several different models such as Denoising AutoEncoder, tensor decomposition, word2vec or spectral embeddings. Experimental results show the effectiveness of these models in several different datasets.

The author uses several different models (both recent studies like Node2Vec / sen2vec, and older results like spectral or tensor decomposition). The idea of studying embedding of a hypergraph is interesting and novel, and the results show that several different kinds of methods can all provide meaningful results for realistic applications.

Despite the novel idea about hyperedge embedding generation, the paper is not easy to follow.
The introduction of ``hypergraph`` takes too much spapce in preliminary, while the problem for generating embeddings of hyperedge is the key of paper.

Further, the experiments only present several models this paper described.
Some recent papers about hypergraph and graph structure (even though cannot generate embeddings directly) are still worth mention and compare in the experimental section. It will be persuasive to mention related methods in similar tasks.

it would better better if the author can add some related work about hyperedge graph studies.

---

> ### Author Response · Authors · 2018-01-05
> **Thanks for the reviews**
>
> Dear Reviewer,
>
> > This paper studies the problem of representation learning in hyperedges. The author claims their novelty for using several different models to build hyperedge representations. To generate representations for hyperedge, this paper proposes to use several different models such as Denoising AutoEncoder, tensor decomposition, word2vec or spectral embeddings. Experimental results show the effectiveness of these models in several different datasets.
>
> > The author uses several different models (both recent studies like Node2Vec / sen2vec, and older results like spectral or tensor decomposition). The idea of studying embedding of a hypergraph is interesting and novel, and the results show that several different kinds of methods can all provide meaningful results for realistic applications.
>
> > Despite the novel idea about hyperedge embedding generation, the paper is not easy to follow. The introduction of ``hypergraph`` takes too much space in preliminary, while the problem for generating embeddings of hyperedge is the key of paper.
>
> Ø We have taken care these.
>
> > Further, the experiments only present several models this paper described. Some recent papers about hypergraph and graph structure (even though cannot generate embeddings directly) are still worth mention and compare in the experimental section. It will be persuasive to mention related methods in similar tasks. it would better better if the author can add some related work about hyperedge graph studies.
>
>  Ø We have revised our related work.

---

### Official Review · AnonReviewer2 · 2017-11-30
**The methods presented in the paper are direct adaptation of existing techniques and rely on heursitcs to work. These methods need to be more thoroughly evaluated (among themselves, to know which method suits for a given problem) as well as against against a simple baseline.**

**Rating:** 5
**Confidence:** 4

**Review:**

This paper addresses the problem of embedding sets into a finite dimensional vector space where the sets have the structure that they are hyper-edges of a hyper graph. It presents a collection of methods for solving this problem and most of these methods are only adaptation of existing techniques to the hypergraph setting. The only novelty I find is in applying node2vec (an existing technique) on the dual of the hypergraph to get an embedding for hyperedges.

For several methods proposed, they have to rely on unexplained heuristics (or graph approximations) for the adaptation to work.  For example, why taking average line 9 Algorithm 1 solves problem (5) with an additional constraint that \mathbf{U}s are same? Problem 5 is also not clearly defined: why is there superscript $k$ on the optimization variable when the objective is sum over all degrees $k$?

It is not clear why it makes sense to adapt sen2vec (where sequence matters) for the problem of embedding hyperedges (which is just a set). To get a sequence independent embedding, they again have to rely on heuristics.

Overall, the paper only tries to use all the techniques developed for learning on hypergraphs (e.g., tensor decomposition for k-uniform hypergraphs, approximating a hypergraph with a clique graph etc.) to develop the embedding methods for hyperedges. It also does not show/discuss which method is more suitable to a given setting. In the experiments, they show very similar results for all methods. Comparison of proposed methods against a baseline is missing.

---

> ### Author Response · Authors · 2018-01-05
> **Agree to the confusion over baselines**
>
> Dear Reviewer,
>
> Thanks for you valuable comments, we have responded to them in-line below.
>
> > This paper addresses the problem of embedding sets into a finite dimensional vector space where the sets have the structure that they are hyper-edges of a hyper graph. It presents a collection of methods for solving this problem and most of these methods are only adaptation of existing techniques to the hypergraph setting.
>
> Ø   We agree our paper lacked a comprehensive clarification of the baseline methods, leading to some confusion. We have revised the paper with this clarification. Specifically, we propose two methods: hypergraph tensor decomposition and hypergraph auto-encoder, as our main contributions. Both these methods are designed to take into account the hypergraph structure in a principled manner. Rest all the methods are adaptations of existing graph or language models which have to be adapted by use of proxy or heuristics as they are not designed for hypergraphs. In this sense, our proposed techniques are more general.
>
> > The only novelty I find is in applying node2vec (an existing technique) on the dual of the hypergraph to get an embedding for hyperedges.
>
> Ø   Regarding novelty aspect, we have clearly listed the novelties we claim in the paper’s introduction, which we again comprehend as follows:
> o   We propose the concept of dual tensor, which is itself novel and allows us to get hyperedge embedding directly.
> o   Our proposed hypergraph tensor decomposition method is designed for general hypergraphs (containing different cardinality hyperedges). Therefore, this tensor decomposition is different than simple uniform hypergraph tensor decomposition which is restricted to fixed cardinality hyperedges (i.e. uniform hypergraph).
> o   Use of de-noising auto-encoder in a hypergraph setting is novel. The idea of creating noise using random-walks over hasse diagram topology is original and unique.
>
> Apart from the methods we propose, we have used several interesting tricks and heuristics in our baselines while adapting them for hypergraph setting.
> o   Use of node2vec over hypergraph dual. (Reviewer has pointed this out himself)
>
> o   Using hyperedges to model sentences is a novel idea and opens up possibilities of various applications using higher order topological methods for modeling language structure. We show one possible application.
>
> o   Adapting set structured data to fit in a sequence based language model using proxy text is an interesting idea.
>
> > For several methods proposed, they have to rely on unexplained heuristics (or graph approximations) for the adaptation to work. For example, why taking average line 9 Algorithm 1 solves problem (5) with an additional constraint that \mathbf{U}s are same?
>
> Ø  Although its a heuristic, but in our implementation we empirically observe that our algorithm converges successfully. Averaging can be interpreted as equal contribution from the latent factors learned from different cardinality (uniform) sub-hypergraphs. Also the optimization objective of problem (5) is unweighted.
>
> > Problem 5 is also not clearly defined: why is there superscript $k$ on the optimization variable when the objective is sum over all degrees $k$?
>
> Ø  We have clarified the problem definition more precisely.
>
> > It is not clear why it makes sense to adapt sen2vec (where sequence matters) for the problem of embedding hyperedges (which is just a set). To get a sequence independent embedding, they again have to rely on heuristics.
>
> Ø  As clarified above, hyperedge2vec using sen2vec is a baseline method. Given sen2vec is for sequences, we have to generate proxy node sequence i.e. proxy text, to be used as input for the sen2vec.
>
> > Overall, the paper only tries to use all the techniques developed for learning on hypergraphs (e.g., tensor decomposition for k-uniform hypergraphs, approximating a hypergraph with a clique graph etc.) to develop the embedding methods for hyperedges. It also does not show/discuss which method is more suitable to a given setting. In the experiments, they show very similar results for all methods. Comparison of proposed methods against a baseline is missing.
>
> Ø  As pointed out previously, over all we propose two methods which are principally designed to handle hypergraph structured data. Our experiments show that accuracy wise our methods perform similar to those of baselines which are not designed for hypergraphs. Moreover, our tensor based method is quiet efficient as compared to deep-learning based auto-encoder method. We therefore, argue that we have proposed more general methods which are suited for hypergraphs (and therefore also for graphs) while maintaining accuracy and efficiency.

---

### Decision · Program_Chairs · 2018-01-29
**ICLR 2018 Conference Acceptance Decision**

**Decision:**

Reject

**Comment:**

While there are some interesting and novel aspects in this paper, none of the reviewers recommends acceptance.